# Mass Transfer in the Processes of Native Lignin Oxidation into Vanillin via Oxygen

Valery E. Tarabanko [1,*], Konstantin L. Kaygorodov [1], Aleksandr S. Kazachenko [1,*], Marina A. Smirnova [1], Yulia V. Chelbina [1], Yury Kosivtsov [2] and Viktor A. Golubkov [1]

[1] Institute of Chemistry and Chemical Technology SB RAS, FRC "Krasnoyarsk Science Center SB RAS", 50/24, Akademgorodok, 660036 Krasnoyarsk, Russia; kulik@icct.ru (K.L.K.); smirnova.ma@icct.krasn.ru (M.A.S.); agafon5@mail.ru (Y.V.C.); golubkov.va@icct.krasn.ru (V.A.G.)

[2] Department of Chemical Technology, Tver State Technical University, 22, nab. A. Nikitina, 170026 Tver, Russia

[*] Correspondence: veta@icct.ru (V.E.T.); leo_lion_leo@mail.ru (A.S.K.)

**Abstract:** The influence of mass transfer intensity on the kinetics of the catalytic oxidation of flax shives with oxygen in alkaline media to aromatic aldehydes and pulp was studied. The process was carried out in two autoclaves, with moderate stirring (stirrer engine of 8 W) and intense stirring (stirrer engine of 200 W). The oxidation of flax shives into vanillin, syringaldehyde, and pulp was shown to proceed as a completely diffusion-controlled process under the studied conditions, both moderate and intense stirring. Depending on the process conditions, it can be limited by stages of oxygen transfer through the diffusion boundary layer near the gas–liquid interface (low intensity of mass transfer) as well as by reagents' inner diffusion in the porous and solid matter of the flax shive particle (high intensity of mass transfer). The results on the influence of the stirring speed and volume of the reaction mass on the rates of oxygen consumption and vanillin accumulation were obtained. They were described using a known simple model connecting the intensity of mass transfer and the stirring power density in the bulk of the liquid phase in terms of algebra equations.

**Keywords:** oxidation; lignin; vanillin; *Linum usitatissimum*; flax shives; mass transfer; diffusion; kinetics

## 1. Introduction

Processing agricultural and wooden wastes into valuable chemicals is an actively developing field of research. It should be noted that a significant part of these feedstocks is waste from the mechanical processing of grassy plants and timber. Lignins in such materials have the structure and chemical reactivity of native ones, as opposed to, for example, condensed Kraft lignins produced via sulfate pulping. This property offers the opportunity to obtain high yields of monophenolic products via the oxidation or hydrogenation of lignin-containing agricultural wastes [1–3].

There is a great variety of agricultural wastes, and one of them is flax shives. They are produced (up to 70 wt.% of raw flax) during the processing of the plant into cellulose fibers and are characterized by high lignin content (20–28%) and good yield (up to 20%) of aromatic aldehydes, e.g., vanillin and syringaldehyde, in oxidation via oxygen or nitrobenzene [2,4,5], also performing well in hydrogenolysis and pyrolysis [6,7].

Although the processes of lignin oxidation into the aromatic aldehydes have a long history (50–70 years [3,8–12]), the influence of diffusion and mass transfer intensity on the reaction rate and selectivity has not been systematically studied. Often, authors who study the kinetics of these processes assume that they proceed under a chemically controlled kinetic mode [13–15]. It was once shown that in the oxidation of pine wood, vanillin yield depended on stirring speed [16], meaning that the process proceeds in a diffusion-controlled mode. Pacek showed that increasing the stirring speed leads to higher rates of oxygen consumption and vanillin accumulation, which also demonstrates diffusion

limiting in at least some of the experiments on the oxidation of lignosulfonates [17]. In one paper [18], a transition from diffusion to chemical control was observed after an increase in mixing intensity during the oxidation of powdered aspen wood at 110 °C.

While studying mass transfer in the process of lignocellulose oxidation via oxygen, several complications and difficulties should be taken into account.

Irregular structures of lignins are formed as a result of the oxidative dehydrogenation of coniferyl, synapyl, and p-coumaric alcohols. Lignins are kinetically inhomogeneous during their oxidation into vanillin, and in the first approximation it may be considered as consisting of two fractions (Figure 1) [3]. The first contains more β-O-4 bonds between coniferyl units (structure (**I**)), and the second fraction (**II**) includes more 5-5′ bonds. The first structure can produce vanillin during oxidation, while the second cannot.

**Figure 1.** The most important structures of coniferous lignin containing β-O-4 (**I**) and 5-5′ (**II**) bonds).

The process of lignin oxidation includes four phases: liquid water solution of alkali, gaseous oxygen, catalyst, and solid particles of flax shives. The latter solid particles change their size and reactivity during the oxidation process, starting from a 1 mm size down to micronized pulp particles, and this is a topochemical process with unknown regularities. Once more, lignin is an irregular polymer (Figure 1) that is composed of different structures with different activities.

The first stage of the process involves the delignification, destruction, and dissolution of solid lignin as a result of its heterogeneous reaction with alkali and solubilized oxygen (Figure 2). The internal diffusion of reagents and products of this stage should be taken into account while discussing its role in the process in total. Probably, oxygen consumption in this stage is relatively slow compared to the total oxygen consumption in the process.

It should be noted that there is a zero stage of the process: dissolution of the most reactive part of lignin in the alkaline solution during the heating of the reactor without oxygen. A part of vanillin is formed in this stage as a result of the retroaldol reaction of substituted coniferyl aldehyde. According to our viewpoint, all vanillin is obtained due to such a retroaldol reaction, and oxygen is necessary in this process to oxidize substituted coniferyl alcohols into aldehydes.

The second (and the main) stage of the process is the catalytic and non-catalytic oxidation of oligomers of destructed lignin. The main part of vanillin is obtained in this stage, and the highest proportion of oxygen is consumed here. The total oxygen consumption exceeds the vanillin yield (mol per mol) in the process via a factor of 20–30 due to the deep oxidation of lignin and soluble carbohydrates in the solution. The obtained carboxylic acids and $CO_2$ consume analogous quantity of alkali.

The third (and undesirable) stage is the oxidation of vanillin into vanillic acid and, finally, $CO_2$ and $Na_2CO_3$. Vanillin, as a phenol, is relatively more stable compared to

phenolic groups of lignin under the conditions of oxidation due to the electron-accepting properties of the carbonyl p-substituent.

All three consecutive stages, except the zero one, occur simultaneously, and to separate the kinetics of a certain stage is a special and complicated task.

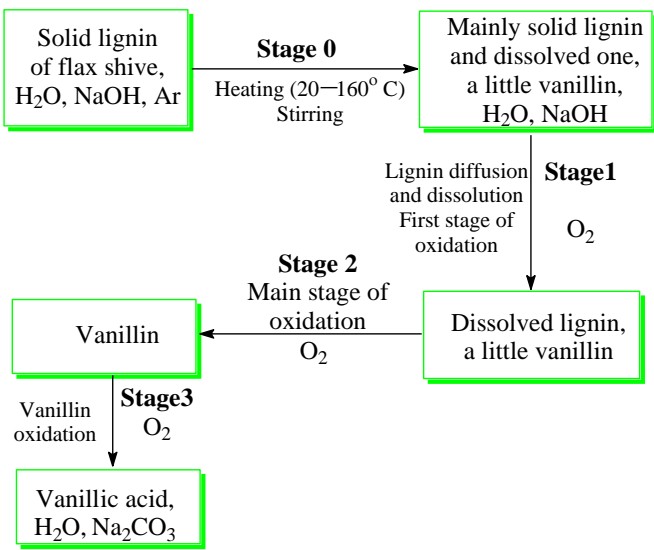

**Figure 2.** Main stages of the process of lignin oxidation.

The authors of [19,20] proposed the first quantitative model for the influence of mass transfer intensity on the rates of oxygen consumption and vanillin accumulation. This study was complicated by the presence of mucilage in the flax shives and via the formation of gels in the aqueous phase. This complication was mostly overcome by acid prehydrolysis of the shives [19]. It was shown that the oxidation of the shives at 160 °C is described via linear dependences of the oxygen consumption and the vanillin accumulation rate logarithms versus the logarithm of the stirring speed:

$$lnW_{O_2} = 1.87 \, lnN + a, \tag{1}$$

$$lnW_V = 1.86 \, lnN + b, \tag{2}$$

where $W_{O_2}$ and $W_V$ are the rates of oxygen consumption and vanillin accumulation, respectively (mol/min), and $N$ is the stirrer rotation speed ($min^{-1}$). These dependences coincide with the earlier-obtained and recommended models for the description of mass transfer in gas–liquid systems [21,22]. These models explain the overall process rate limitation via the rate of oxygen diffusion through the liquid near-surface region at the gas–liquid phase boundary and are often interpreted as examples of a more general model that links the rate of a diffusion-limited chemical process to stirring power density [23].

Discussions of mass transfer can be facilitated using a schematic that illustrates mass flows in systems involving a "gas-liquid-solid catalyst", for example, offered in [24]. Figure 1 presents this schematic in the form adapted for the oxidation of flax shives into vanillin and cellulose. Momentarily omitting the catalyst (the role of which will be discussed in a later section), the reaction environment consists of three phases: aqueous liquid, gas (oxygen, argon, and water vapor), and solid (porous shives particles). At the gas–liquid and the solid–liquid phase boundaries, there are diffusion boundary layers, and transferring the reactants and products through them may limit the rate of the chemical process.

On a qualitative level, the oxygen concentration gradient in the diffusion layer at the gas–liquid boundary is quite obvious (Figure 3).

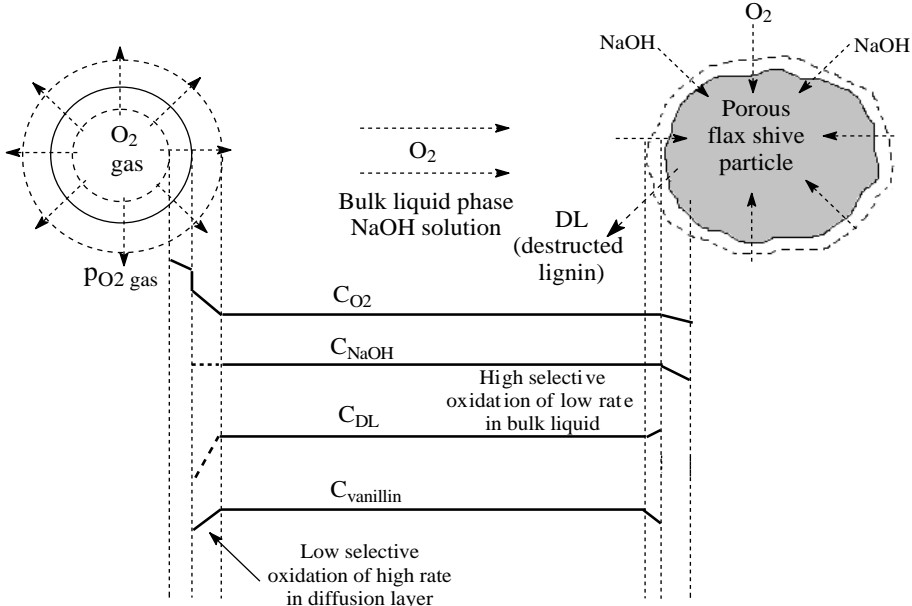

**Figure 3.** Schematic of mass transfer, chemical reactions, and concentration profiles for the principal reactants and products in the oxidation of flax shives via oxygen into vanillin and cellulose (stirring engine of 8 W).

The concentration gradient of dissolved oxygen in the liquid bulk is, in many cases, practically zero, and when we hereinafter mention the homogeneous liquid phase, this is the meaning we apply to it. The depicted concentration gradients of the reactants and the products at the solid–liquid phase boundary are derived from experimental data, and will be discussed in a later section. In terms of Figure 3, the dependences of the rates of oxygen consumption (1) and vanillin accumulation (2) versus stirring speed can be most naturally explained by overall limiting of the process via transfer of oxygen through the liquid diffusion layer at the gas–liquid phase boundary. The thickness of this layer decreases under a faster stirring speed, and this determines the observed influence of the mass transfer intensity on the oxygen consumption and vanillin accumulation rates.

When discussing the rate-limiting stage of a chemical process, a clarification of this term is necessary. For a homogeneous chemical process proceeding as a sequence of stages with the same kinetic order (e.g., first order), the limiting stage is determined using the lowest rate constant. This simplest case is described in many textbooks on chemical kinetics. A more complicated situation is a homogeneous chemical process composed of stages with different kinetic orders. Direct comparisons of rate constants become impossible; the rates ($mol\,L^{-1}\,min^{-1}$) of all stages after the limiting one become equal to that of the latter, just like in the previously described situation. To determine the limiting stage in complex systems (including heterogeneous ones, like the process that we presently study), the concept of a stage's characteristic time can be used: the rate-limiting stage has a considerably higher characteristic time compared to the others [24]. Another noteworthy property of heterogeneous kinetics is that under chemical rate control, reactant concentrations in diffusion layers at phase boundaries are zero.

An important property of diffusion-limited processes with stirring is that their rates may not depend on the reaction mixture volume. This independence is important for the minimization of power consumption for stirring. Also, it can most obviously demonstrate the validity of the model for the control of the process rate via the stirring power density [21,22]. Aside than one publication [19], there exist no studies on the influence of the reaction mixture volume on the rate of lignin oxidation. In that work, it was shown that at a constant stirring speed, the rate of oxygen consumption ($mol\,min^{-1}$) does not depend (in a certain range) on the volume of the reaction mixture with fixed reactant concentrations. This result corresponds to the model of the overall rate control via the stirring power

density [21,22]. However, the literature has not revealed any discussions on the influence of the reaction mixture volume on the yield of the aromatic aldehydes from the processes of lignin oxidation.

Thermodynamically, during lignin oxidation, vanillin and syringaldehyde are obviously intermediate products, and estimates of their degradation rates were given several times [25–29]. Selectivity of processes under diffusion rate control is generally lower than under chemical control, and under full diffusion control, it is ultimately possible that the aromatic aldehydes are completely oxidized into more stable products within the liquid diffusion layer at the gas–liquid boundary.

However, there is a report of a 28% vanillin yield from pine lignin [27], which matches the effectiveness of oxidation via nitrobenzene [8]; this result was obtained under diffusion control in the same conditions as those employed by the authors of [19]. Vanillin yield from nitrobenzene oxidation is generally accepted as the theoretical maximum for the oxidation of lignins via oxygen under the chemical rate control mode. The vanillin and syringaldehyde yields from diffusion-controlled oxidation of flax shives via oxygen are not significantly lower (17–30 relative percent) than from the nitrobenzene process [19]. This raises the question about the reasons for the high selectivity of diffusion-controlled oxidation of lignins via oxygen into aromatic aldehydes. An answer to this question requires knowledge about the influence of the mass transfer intensity on the kinetics and the selectivity of the oxidation of lignins into vanillin and syringaldehyde.

The presented literature analysis allows for the formulation of two interconnected objectives of this paper. Firstly, studying the influence of the reaction mixture volume on yields of the primary products from the oxidation of lignins into the aromatic aldehydes, exemplified by flax shives. Secondly, studying individual stages of the mass transfer involved in the oxidation of flax shives into vanillin and syringaldehyde to explain the obtained kinetic results. Based on the combined results, the influence of mass transfer conditions on the oxidation process selectivity will be discussed.

## 2. Results and Discussion

In order to fulfil the formulated objectives, the following aspects of the process were studied:

1. Influence of the reaction mixture volume on the dynamics of oxygen consumption and the formation of the principal products during the oxidation of mucilage-free flax shives. Absence of the mucilage in the reaction mass avoids unnecessary complications of the mass transfer.
2. Influence of the reaction mixture volume and the stirring speed on the rate of homogenization of the aqueous phase, and on sorption of alkali via the powdered shives' suspension.
3. Reasons for differing rates of oxygen consumption in the initial (one minute) and main (2–40 min) sections of the kinetic curves.
4. Influence of the stirring speed on the nature of the rate-limiting stage of the process of the oxidation of flax shives into vanillin and cellulose.
5. Comparison of the oxidation processes in reactors with low stirring power density (stirrer engine power: 8 W) and with intense stirring (engine power: 200 W).

All of the hereinafter presented experiments, except in Section 2.5, were conducted in the reactor with the eight watt stirrer.

### 2.1. Influence of the Reaction Mixture Volume on the Process of the Catalytic Oxidation of Flax Shives

Figures 4–6 and Table 1 present experimental data on the primary characteristics of flax oxidation with a fixed stirring speed—oxygen consumption dynamics and its volume corresponding to the maximum vanillin yield, vanillin accumulation dynamics, and its maximum yield, and the yield of lignoacids as a side product—and how they are influenced by the volume of the reaction mixture with a fixed concentration.

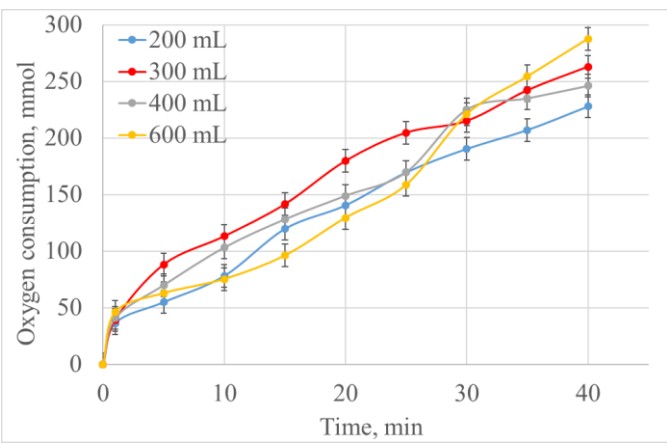

**Figure 4.** Dynamics of oxygen consumption in the oxidation of different volumes (200–600 mL) of the reaction mixture. Flax shives from Belarus 50 g/L; NaOH 50 g/L; oxygen 0.2 MPa; 160 °C; and stirrer rotation rate 500 min$^{-1}$.

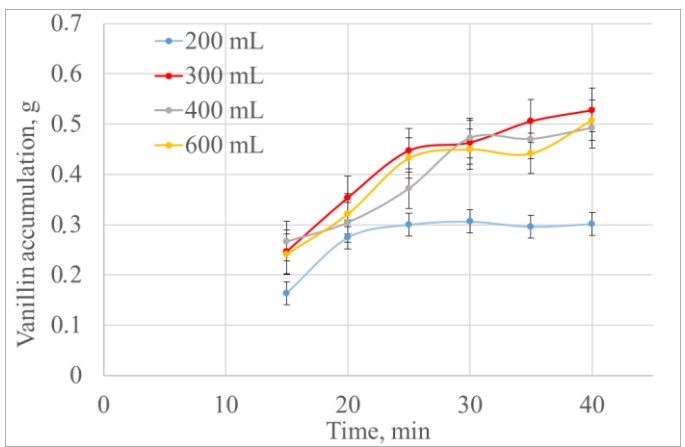

**Figure 5.** Dynamics of vanillin accumulation (in grams) in the oxidation of different volumes (200–600 mL) of the reaction mixture. For the mixture composition, see Figure 4.

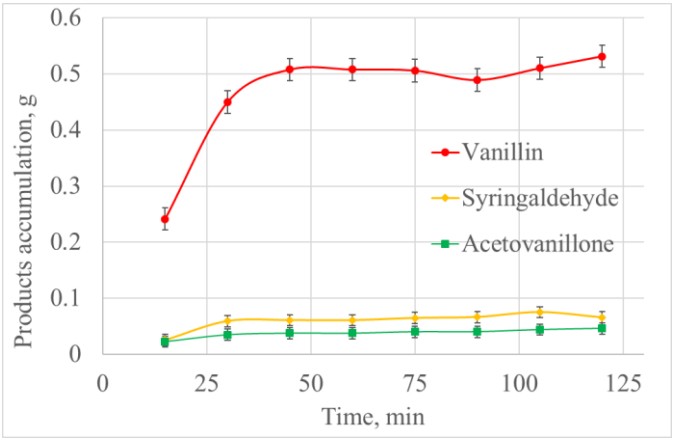

**Figure 6.** Accumulation of the principal products (in grams) in the oxidation of flax shives for the reaction mixture volume 600 mL. For the mixture composition, see Figure 4.

The most important result that follows from these data is that the reaction mixture volume has practically no effect on the oxygen consumption rate (mmol min$^{-1}$) and its integral consumed amount (mmol) (Figure 4, Table 1), as well as on the vanillin and

lignoacid yields (grams) (Table 1). In the volume range of 300–600 mL, these characteristics fluctuated within the of range ±10–15% or less (Table 1).

**Table 1.** Influence of the reaction mixture volume on the amounts of the principal products formed in the oxidation of flax shives. For details of the conditions, see Figure 4.

| Volume (mL) | Oxygen Consumption at 40 min (mmol) | Oxygen Consumption Rate (mmol/min) | Vanillin Yield (g) | Lignoacid Yield (g) | Vanillin Yield Based on the Initial Lignin (wt.%) | Lignoacid Yield Based on the Initial Lignin (wt.%) |
|---|---|---|---|---|---|---|
| 200 | 228 | 4.79 | 0.302 | 1.36 | 12.3 | 49 |
| 300 | 263 | 4.99 | 0.528 | 1.89 | 14.4 | 45 |
| 400 | 246 | 5.06 | 0.492 | 2.2 | 10.0 | 39 |
| 600 | 288 | 6.03 | 0.508 | 2.36 | 6.9 | 28 |
| Average results (300–600 mL) | 266 ± 14 | 5.36 ± 0.45 | 0.51 ± 0.02 | 2.13 ± 0.23 | - | - |

A simple and natural explanation for this result can be found in the model that links the rate of a chemical process limited by mass transfer to the stirring power density [19–23,30,31]. The simplest description of the mass transfer rate in a gas–liquid system (which corresponds to the oxygen consumption rate in our case) is provided by Equations (3) and (4) [31].

According to the chosen model, the transfer coefficient depends on the stirring power density:

$$\frac{dq}{dt} = \beta_{vol} \cdot ([C_{sat.} - C], \text{mol m}^{-3}\,\text{s}^{-1} \tag{3}$$

where $q$ denotes oxygen consumption, $\beta_{vol}$ is the volumetric coefficient of mass transfer ($\text{s}^{-1}$), and $C_{sat.}$ and $C$ are the saturation and current aqueous concentrations of oxygen, respectively. Extrapolation of the data on $O_2$ solubility in 1 M NaOH at 373 K, 1 Bar (0.51 mmol/L) [32], to 160 °C and 2 Bar gives an $O_2$ solubility ($C_{sat}$) of appr. 0.8 mmol/L.

$$\beta_{vol} = const \cdot \left(\frac{P}{V}\right)^a, \tag{4}$$

where $P$ and $V$ denote the stirring power and the reaction mixture volume, respectively, and $a$ is an experimentally determined exponent. For the oxidation of sodium sulfite solutions under a low stirring intensity, $a = 0.83$ was obtained, whereas a high stirring intensity resulted in a lower exponent of $a = 0.39$ [31].

In order to present Equations (3) and (4) in terms of stirring speed, we assumed an exponential dependence between the power and the stirrer rotation rate:

$$P = const \cdot N^b, \tag{5}$$

where $P$ is the stirring power, $N$ is the rotation rate, and $b$ is an experimentally determined coefficient. Combining Equations (4) and (5), Equation (6) can be obtained, which links the volumetric mass transfer coefficient to the stirrer rotation rate and the reaction mixture volume:

$$\beta_{vol} = const \cdot \left(\frac{N^b}{V}\right)^a = const \cdot \frac{N^{ab}}{V^a}, \tag{6}$$

and Equation (7) that combines logarithmic dependences (1), (2), and independence of the integral flow (mol $\text{min}^{-1}$) from the mixture volume when the value of $a$ is near unit:

$$ln(\beta_{vol} \cdot V) = ab \cdot lnN + (1 - a) \cdot lnV + const. \tag{7}$$

Continuing with the discussion on the obtained results, it should be noted that in a previous work, the volume independence of the oxygen consumption rate in the oxidation of flax shives from Tver was observed in a narrower range of 200–350 mL [19] compared to the data presented in Figure 2 (200–600 mL). These differences may be linked to the mucilage contained in the Tver shives, which increases the reaction mass viscosity, therefore complicating the mass transfer (Bingham suspension). The present work used shives from Belarus that contain no mucilage.

A decrease in the peak vanillin concentration with a 200 mL reaction volume compared to the process with a 300 mL volume (Figure 5) can be explained by excessive oxidation of the product. Prior information states that the peak vanillin concentration is attained at a duration of 40 min [19], and the vanillin accumulation curve in a large volume (600 mL) of the reaction mixture corresponds to this duration (Figure 6).

In addition to vanillin and cellulose, the oxidation process produces lignoacids that precipitate as tar when the post-reaction mass is acidified [29]. The data presented in Table 1 show that a three-fold increase in the reaction volume results in a nearly two-fold decrease in the lignoacid yield based on the initial lignin (from 49 to 28 wt.%). One of the conclusions from this fact is that lignoacids form as a result of oxidation. Their formation can also be attributed to alkaline delignification, non-oxidative hydrolysis reactions, but the dependence in Table 1 shows that oxidation increases the lignoacid yield. Its value corresponds to the amount obtained during the oxidation of Kraft lignins: 18–28% in optimal conditions; 78% and above at lower oxidation depths [29]. The obtained lignoacids have a low molecular mass with high dispersity (Mn = 1300–2100, Mw = 3600–8400, and polydispersity index PDI = 2.6–4.0).

Therefore, the presented results show that at a fixed stirring speed, there exists a range of volumes of the reaction mixture ($V_{max}/V_{min}$ = 2–3) in which the overall values of the oxygen consumption rate (mmol min$^{-1}$), vanillin yield (g), and lignoacid yield (g) are determined via the stirring power density, being independent from the reaction volume [19–23]. This means that there exists an even more narrow range of the reaction mixture volume (exemplified in our experiments by the volume 300 mL) in which the vanillin yield based on the initial lignin is the highest, with minimal expenditure of the reactants for its production.

If the rate of the studied process is controlled via the stirring power density, then increasing the concentration of lignocellulosic material in the reaction mass at a fixed stirring intensity may not lead to the proportional increase in the yield of the target products. We observed such facts earlier. A two-fold decrease in powdered birch wood loading into the reactor (from 10 to 5 wt.%) did not change the resulting masses of vanillin and syringaldehyde, whereas their yields based on the initial lignin became twice as high (43 wt.%), which approaches the result from the oxidation via nitrobenzene (47%) [3]. A two-fold increase in powdered aspen wood loading into the reactor (from 3.3% to 6.6%) in the process of its oxidation in alkaline medium had practically no effect on the initial rate of oxygen consumption [18].

The presented results revealed that the maximum yield of the aromatic aldehydes from a given reactant loading in diffusion-limited processes of oxidation via oxygen can only be obtained under a quite narrow range of the reaction mixture volumes and of lignocellulosic material concentration therein. Excessive volumes of the mixture will decrease the yields of vanillin and syringaldehyde based on the initial reactants due to a lower density of stirring power. On the other hand, when the reaction volume is below optimal, the yield may drop as the target products themselves become oxidized.

In the studied process, we can point out several stages of mass transfer that are significant for the analysis of the studied process; the most important ones are the transfer through diffusion layers and the phase boundaries at the gas–liquid and solid–liquid boundaries (the latter being the surface of the shives' particles), diffusion through the bulk of the shives' particles, and the forced convective transfer by stirring of the liquid bulk (Figure 1). We will now discuss our model conditions' study of the dynamics of these stages

of the flax shives' oxidation process into vanillin in order to draw conclusions about the limiting stage of the process.

## 2.2. Influence of the Stirring Speed and the Reaction Mixture Volume on the Liquid Phase Homogenization Rate

Equalization of the concentrations of the dissolved reactants and reaction products across the liquid phase (hereinafter, liquid phase homogenization) is generally considered to be the fastest of the mass transfer stages [17,24,31]. Experimentally, the homogenization rate can be estimated by observing the duration of coloration equalization of a pre-added indicator after a pulsed addition of a small volume of alkaline solution [32].

The obtained results (Figure 7, left) show that the aqueous phase homogenization rate increases with a higher stirring rate and a lower liquid volume. The homogenization rate values are in the range of 0.25–1.35 s$^{-1}$. Random deviations in the measurements of individual homogenization durations were quite high, and the standard error in coefficients of linear fitting was in range of $\pm$20–50%. Similar results are obtained for the equalization of an added acid in alkaline solution with suspended shives (Figure 7, right). Despite a large error of individual measurements, qualitative patterns were readily apparent, and the range of the measured homogenization duration values (0.7–4 s) was vastly shorter than the overall duration of the oxidation process, by three to four orders of magnitude (Figures 4–6).

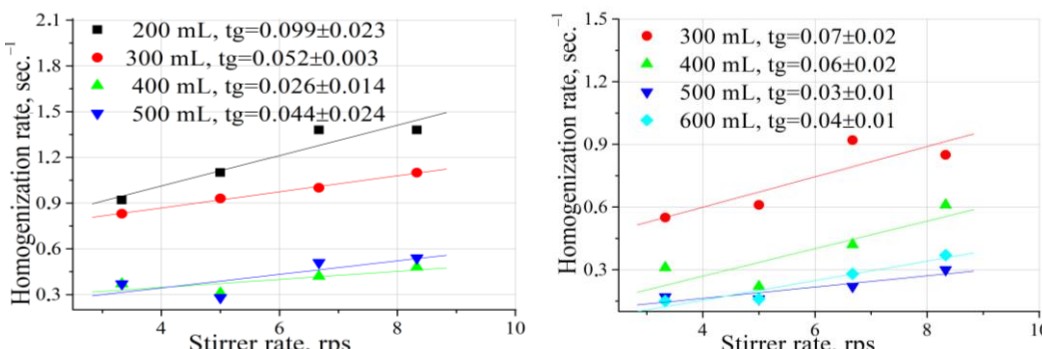

**Figure 7.** Influence of the stirring rate on the aqueous phase homogenization rate. Equalization rate of the concentrations of added alkali (**left**) or acids (**right**). Experiment conditions: suspension of 5 wt.% flax shives in distilled water (**left**) or in pre-mixed 2 mmol/L alkali (**right**).

There is a noteworthy rift between homogenization rates in liquid volumes of 200–300 mL and 400–600 mL. This rate upsurge may have been caused by non-Newtonian behavior of the viscous suspension, where its viscosity increases with lowering gradients of the fluid motion speed at greater distances from the stirrer bar due to low stirring power [19].

The presented results show that the aqueous phase homogenization stage in the process of lignin oxidation into vanillin and syringaldehyde does not limit the overall rate, similar to other processes with stirring. The aqueous phase is homogeneous, i.e., concentrations of the reactants are equal across its entire bulk at any given moment in time (Figure 3).

## 2.3. Influence of the Stirring Speed and the Reaction Mixture Volume on the Rate of Alkali Sorption by Powdered Flax Shives

The last stages of mass transfer in the process of vanillin formation are the diffusion of the reactants and the products through the porous solid body of flax shives. In order to estimate their rates, we used the simplest model process—diffusion of alkali into and within this substrate. Alkaline solution was added to shives suspended in water, and the pH of the aqueous phase was recorded as a function of time. The first stage—equalization of alkali concentration across the liquid bulk—is fast, as was described in the previous

section. Then, significantly slower phenomena take place—alkali diffusion in the capillary structure of the solid shives' particles and (possibly) inside their solid bulk. The eventual acid–base interaction of the hydroxide anion with acidic protons of the lignin is effectively instantaneous on the time scale of the preceding stages. We studied the alkali sorption kinetics at a loading of 0.2 mmol (8 mg) per one gram of the shives. This loading appeared to demonstrate the kinetics most clearly.

Interaction of the alkali with the shives proceeds considerably slower (dozens of minutes, Figure 8) than the homogenization of the aqueous phase (several seconds, Figure 7). Within the duration of the experiment, the measured pH dropped by a unit; hence, the shives adsorbed around 90% of the added alkali. The graphs of pH versus time were linear; therefore, it is possible to interpret them in terms of first-order kinetics, considering how the pH is a linear function of the decimal logarithm of hydroxide ion concentration:

$$pH = -\lg\left[H_3O^+\right], \tag{8}$$

$$pH = -\left(\frac{k}{\ln(10)}\right)t + const, \tag{9}$$

where $\left(\frac{k}{\ln(10)}\right)$ is the slope of the regression line obtained by plotting pH versus time (Figure 8), $k$ is the constant describing the rate of the diffusive transfer of the alkali into the solid particle and inside it, $\ln(10)$ is natural logarithm of 10, and *const* denotes a constant. In first-order kinetics, a reactant's half-life is calculated as $\tau = \frac{\ln(2)}{k}$, and from the data in Figure 6, $\tau$ = 27–29 min can be found.

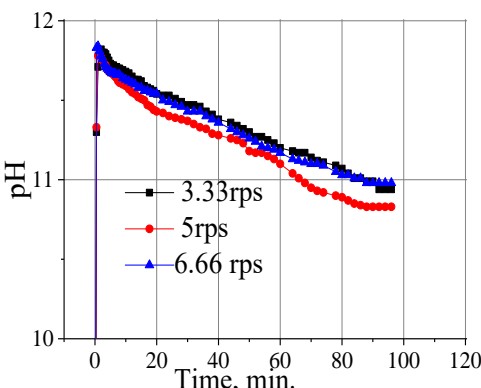 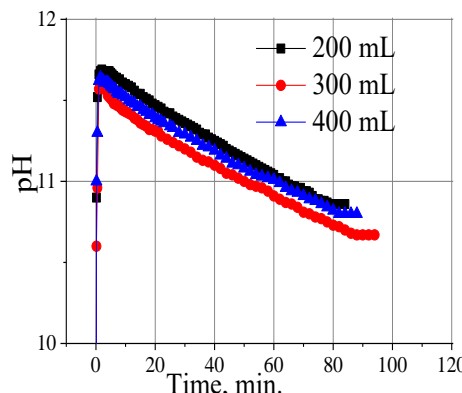

**Figure 8.** Influence of the stirring rate ((**left**), 300 mL of reaction mixture) and the reaction mixture volume ((**right**), 8.33 rps) on the dynamics of pH change following the addition of alkali to a suspension of powdered flax shives (5% wt.) in water (20 °C; 0.2 mmol alkali per flax shive gram).

The apparent rate constant does not depend on the liquid volume nor the stirring rate (Figure 8 and Table 2). This means that the interaction of the shives with the alkali is determined via the diffusion rate inside the shives' particles [30,33], and is not limited via the diffusion of the alkali from the liquid bulk towards the outer surface of the particles. The physical meaning of the calculated rate constant, $k$, can be interpreted as the mass transfer coefficient [24,30,31,33] (see Equation (3)):

$$k = \beta = \frac{D}{\delta}, \tag{10}$$

where $D$ is the molecular diffusivity coefficient, and $\delta$ is the diffusion boundary layer thickness.

**Table 2.** Mass transfer coefficients for the inner diffusion of alkali in the flax shives at different stirring rates and reaction mixture volumes.

| $W_{stirrer}$ (s$^{-1}$) | Reaction Mixture Volume (mL) | k (min$^{-1}$) | $\tau_{1/2}$ (min) |
|---|---|---|---|
| 3.33 | 300 | 0.021 | 32.9 |
| 5 | 300 | 0.026 | 26.5 |
| 6.66 | 300 | 0.023 | 30.0 |
| 8.33 | 300 | 0.025 | 27.6 |
| 8.33 | 200 | 0.024 | 28.75 |
| 8.33 | 300 | 0.025 | 27.6 |
| 8.33 | 400 | 0.024 | 28.75 |

The obtained values of the reactant's half-life (27–29 min), i.e., the characteristic time of the internal diffusion, are close to the overall duration of the studied process (20–40 min; Figures 4 and 5 in the present work; data in the literature [19]). Despite the conditions of this model experiment being far from the practical oxidation process, qualitative comparisons of their durations are possible.

Close rates of the overall oxidation process at 160 °C and of the reactant transfer via internal diffusion in the shives at 20 °C suggest the possibility of the process being limited via the intra-particle diffusion of soluble fragments from lignin depolymerization. Still, the overall rate of flax shive oxidation strongly depends on the stirring intensity (being nearly proportional to stirring rate squared [19]), whereas the internal diffusion rate, by its nature, has no such dependence (Table 2). Therefore, the internal diffusion stages do not actually limit the process of flax shive oxidation under the studied conditions. And so, within the accepted schematic of mass transfer in the oxidation process (Figure 1), only one possibility for the limiting stage is left: oxygen transfer through the diffusion layers at the gas–liquid phase boundary. The internal diffusion proceeds much slower than the aqueous phase homogenization, but the former is still quicker than the oxygen transfer into the liquid bulk (Figure 3).

There are two possible sub-stages in the internal diffusion of reactants: diffusion in the liquid phase within the capillary structure of the flax shives, and diffusion in their solid phase. The feasibility of the former is quite obvious. Now, we will assess the possibility of the latter. According to our experiments (Figure 8), all the initial alkali is absorbed; therefore, the ion exchange capacity of the shives is at least 0.2 mmol g$^{-1}$. Assuming that this minimum estimate of the exchange capacity fully depends on the acidic groups of the lignin (content 29.5 wt.%), pure lignin has an ion exchange capacity of at least 0.2/0.295 = 0.68 mmol g$^{-1}$. The latter value is only twice lower than the capacity of the gelular cationite KU-2 (1.8 mmol g$^{-1}$), whose entire volume participates in the exchange. This comparison suggests that the internal surface of the flax shives cannot afford this ion exchange capacity; therefore, the alkali diffusion and chemisorption occurs inside the solid phase, which potentially undergoes swelling. During the processing at 160–180 °C, which involves the delignification and disruption of the solid phase, this would not be surprising. In the presented experiments at 20 °C, there is no delignification, and only diffusion is responsible for the alkali chemisorption via the solid phase of the shives.

The principal conclusion of this section is that at the low stirring power density used in our work (stirrer engine power: 8 W), the internal diffusion of the reactants and the products inside the flax shive particles proceeds at rates comparable to the overall process rate, but this stage is not the rate-limiting one. External diffusion at the outer surface of the solid particles also does not limit the process rate. In the kinetic schematic of the flax shive oxidation by oxygen, these stages take place after the limiting one, and therefore have the rate of the latter (transfer of oxygen through the gas–liquid surface).

### 2.4. Mass Transfer near the Catalytic Surface

This aspect has been briefly discussed in our paper [20]. Copper sulfate that is loaded into the reactor turns into the hydroxide in the alkaline medium and is then dehydrated into the oxide at the oxidation temperature. The catalyst increases the rate of vanillin accumulation by approximately a factor of 1.5 [20]. XRD revealed that an increase in the stirring rate causes a change in the catalyst phase composition from predominantly $Cu_2O$ (400 rpm) to CuO (700 rpm). This means that despite a considerable size (2–5 micrometer) of the oxide particles observed with SEM and XRD, the composition transition CuO $\leftrightarrow$ $Cu_2O$ not only takes place on the surface but also in the catalyst particle bulk.

### 2.5. Diffusion Limitation in Oxygen Transfer through the Gas–Liquid Phase Boundary

We showed earlier that under the studied conditions in the main section of the kinetic curves (after the first minute of the oxidation process) in the stirring rate range of 100–700 $min^{-1}$, the process is diffusion-limited, and the dependences on the rates of oxygen consumption and vanillin formation versus the stirring rate conform to Equations (1) and (2). The maximum oxygen consumption rate during the main section was 0.5 mmol $L^{-1}$ $s^{-1}$ [19], which is two orders of magnitude greater than the rate of Kraft lignin oxidation under similar conditions that have been described in the literature ($3 \cdot 10^{-3}$ mmol $L^{-1}$ $s^{-1}$) [29], but at the same time, it is lower than the maximum rate of oxygen transfer through the gas–liquid phase boundary, as calculated in the same paper (9 mmol $L^{-1}$ $s^{-1}$) [29].

The main section of the oxygen consumption curve (1–15 min and later) is preceded by a previously undiscussed short initial section shorter than 1 min, during which the oxygen consumption rate is considerably higher (around 2 mmol $L^{-1}$ $s^{-1}$) than during the main one (Figures 4 and 9; Table 3). This situation seems paradoxical; mass transfer limits the oxygen consumption rate during the main section, but there is the initial section where the consumption rate exceeds this diffusive limit.

However, the diffusion-limited oxygen consumption rate is affected by the concentration and activity of reducing agents in the liquid phase [24,31], as such agents can react with oxygen in the diffusion boundary layer, thereby increasing the latter's concentration gradient, ultimately accelerating its diffusion. In fact, in the process of flax shive oxidation, the presence of extremely reactive reductants (e.g., ascorbic acid and 1,2,4-triacetoxy benzene) increases the initial rate of oxygen consumption by a factor of at least 2–2.5 (to 4.6–5.5 mmol $L^{-1}$ $s^{-1}$), while having practically no effect on the consumption rate during the main section of the kinetic curve (1–10 min; Figure 9; Table 3). In the context of this result, the high initial oxygen consumption rate in the process without the added reductants can be explained by the presence of dissolved alkaline lignin that accumulates in the aqueous phase during the heating of the reaction mass water–alkali–shives catalyst without oxygen.

The considerable decrease in the oxygen consumption rate upon the transition from the initial to the main section of the kinetic curve (4–10-fold; Table 3) suggests the following assumption: The concentration of oxygen at the boundary of the liquid diffusion layer—therefore, likewise in the liquid bulk—is high enough that the oxidation during the main section predominantly proceeds in the liquid bulk. In this situation, oxygen and low molecular weight fragments of the lignin macromolecules react in the homogeneous aqueous solution, entering it via diffusion boundary layers. In other words, this crucial chemical stage of the process of flax shive oxidation via oxygen takes place in a homogeneous medium, i.e., without limitation by diffusion and the selectivity problems associated with such a limitation.

**Table 3.** Oxygen consumption (mmol) and its rate (mmol $L^{-1}$ $s^{-1}$) in the process of flax shive oxidation with additional reductants, e.g., ascorbic acid (16.2 g) and 1,2,4-triacetoxy benzene (15 g). For other conditions, see Figure 4.

| t, min | Flax Shive | Ascorbic Acid + Flax shive | Pyrogallol A + Flax Shive | Pyrogallol A |
|---|---|---|---|---|
| | Oxygen Consumption, mmol (Rate of Consumption, mmol/L·s) | | | |
| 0 | 0 | 0 | 0 | 0 |
| 1 | 38.3 (2.1) | 99.9 (5.5) | 83.2 (4.6) | 58.2 (3.2) |
| 5 | 88.2 | 143.2 | 112.4 | 59.8 (0) |
| 10 | 112.7 (0.46) | 165.7 (0.41) | 142.77 (0.6) | 61.1 (0) |

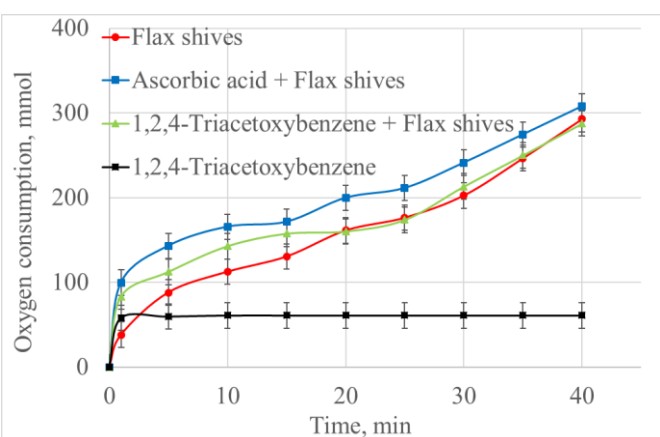

**Figure 9.** Influence of added ascorbic acid and 1,2,4-triacetoxy benzene on the oxygen consumption kinetics during the oxidation of flax shives. For conditions, see Table 3 and Figure 4.

### 2.6. Oxidation with Intensified Mass Transfer

The hereinabove presented results were obtained under a low stirring power density (engine power: 8 W). It was concluded that under these conditions, the overall process rate is limited by the transfer of oxygen from the gas phase into the bulk liquid. Naturally, a question arises: is it possible to accelerate this process by intensifying the mass transfer (potentially excluding the transfer of oxygen into the liquid bulk as the rate-limiting stage)? Experiments with a propeller stirrer (engine power: 200 W) were carried out.

Several concepts regarding the comparison of the mass transfer intensities in the two reactors with the different stirrers can be gained from the data presented in Figure 10, showing how the stirring rate affects the vortex edge height (left) and the diameter of the dry spot at the vortex bottom (right). With the low-power magnetic stirrer, the same vortex shape was obtained at twice the rotational rate compared to the high-power propeller stirrer. The exact power transferred to the liquid was not measured, but in this situation, the ratio of the actual agitation powers between the two stirrers was probably much lower than the ratio of their nominal power values.

Figure 11 depicts the dynamics of oxygen consumption during the shive oxidation process for different rotation rates of the propeller stirrer (200–1200 $min^{-1}$), and the linear logarithmic dependence of the consumption rate versus the stirring rate, similarly to the data presented in the literature [19], was obtained:

$$lnW_{O2} = 1.12 \, lnN - 4.80. \tag{11}$$

This dependence has a lower slope (tg($\varphi$) = 1.12 ± 0.13) compared to the low-power stirrer (tg($\varphi$) = 1.88 [19]). Under chemical or internal diffusion control of the process rate, the latter would not depend on the stirring rate (tg($\varphi$) = 0). Therefore, the decreasing slope indicates a gradual transition from the process being limited by the external diffusion towards the process with internal diffusion control as the stirring power increases.

Our previously obtained results [19] and the data in Figures 4 and 5 and Table 1 indicate that the kinetic patterns of the process conform to the model based on the stirring power density (Equations (4) and (7)). The linear dependences of the oxygen consumption rate versus the stirring rate in logarithmic coordinates (Equation (11) and [19]) also conform to this model.

The time at which the maximum vanillin concentration is attained in the intensely stirred reactor at 1200 min$^{-1}$ (25–30 min, Figure 12) is close to the corresponding time in the reactor with a low stirring intensity at 500 min$^{-1}$ (30–40 min; Figures 5 and 12). A comparison of how the stirring intensity affects the rates of oxygen consumption and vanillin formation showed that the latter was limited by internal diffusion with strong stirring (diffusion of primarily the oligomeric products of lignin destruction), whereas with weak stirring, there was a limitation of vanillin formation via the oxygen transfer through the diffusion layer at the gas–liquid boundary. For these reasons, a higher stirring intensity can elevate the vanillin accumulation rate when the stirring is still sufficiently weak, but eventually, stronger stirring eliminates this dependence.

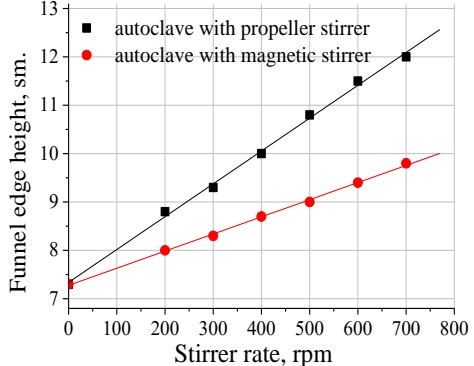 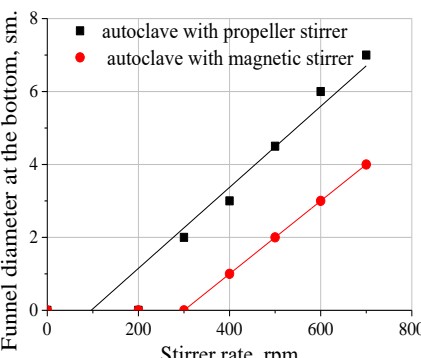

**Figure 10.** Dependence on the vortex shape versus the rotation rate (rpm) of the magnetic stirrer (8 W) and the propeller stirrer (200 W). Suspension of 15 g of flax shives in 300 mL of water.

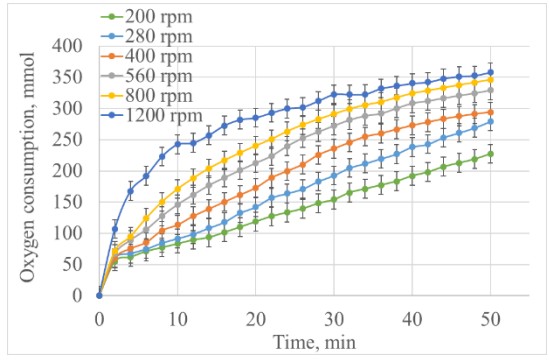

**Figure 11.** Dynamics of oxygen consumption during the flax shive oxidation process for different stirring rates under the conditions of intense mass transfer (propeller stirrer; engine power: 200 W).

The initial rate of oxygen consumption at a stirring rate of 1200 min$^{-1}$ (29 mmol min$^{-1}$, Figure 11) is five times higher than the oxidation rate in the reactor with a low mass transfer intensity (5.3 mmol min$^{-1}$, Figure 4). The oxygen amounts consumed within 30 min in these two experiments (200–250 mmol) were very close due to the gradual deceleration of

the oxidation in the former case. This rate decrease was obviously caused by the decreasing concentration of reductants in the solution as a result of lignin oxidation.

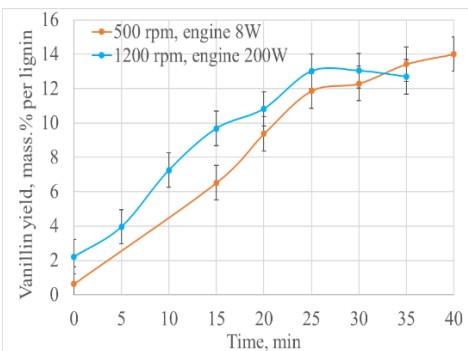

**Figure 12.** Dependence on the accumulation of vanillin versus the process duration under the conditions of intense mass transfer (blue line; propeller stirrer; engine power: 200 W) and low-intensity mass transfer (red line; magnet stirrer; engine power: 8 W).

Maximum vanillin yields based on the initial lignin in both of the modes of mass transfer intensity are practically the same (12–14%, Figures 5 and 12). Therefore, efficient oxidation of native lignins into their aromatic aldehydes is possible under diffusion-limited rates of oxygen transfer through the liquid diffusion layer at the gas–liquid phase boundary, with a low-energy expenditure for stirring.

To summarize, Figure 13 depicts the profiles of the reagent and product concentrations depending on the mass transfer intensity in the process under study. Obviously, the main changes were observed in the diffusion boundary layers while increasing the mass transfer intensity. All the concentration gradients of the reactants and products were close to zero at a high mass transfer intensity. While decreasing it, the concentrations of the products in the diffusion layer at the gas–liquid phase boundary also decrease, but nevertheless, due to the very low volume of the diffusion boundary layers, vanillin yields may attain a theoretically maximum limit that coincides with the efficiency of the nitrobenzene oxidation process [27].

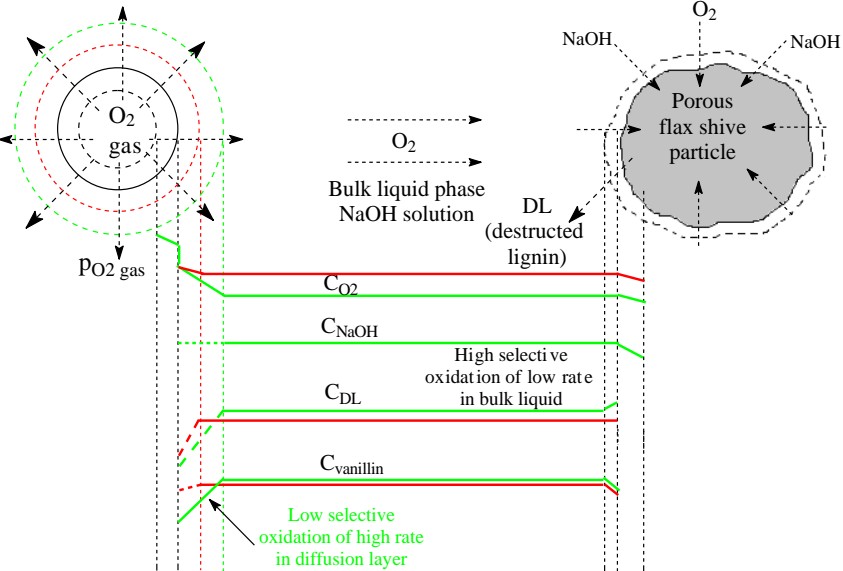

**Figure 13.** Schematic of the mass transfer processes, chemical reactions, and concentration profiles of the principal reactants and products in the oxidation of flax shives via oxygen into vanillin and cellulose, with a high intensity of mass transfer (red lines; stirrer engine 200 W) and a low one (green lines; stirrer engine 8 W).

## 3. Materials and Methods

Air-dried powdered (≤1 mm) flax shives (*Linum usitatissimum*, Rosinka cultivar [19]; harvest of 2021; Belarus origin) were employed for the experiments. Lignin and cellulose content were determined using the standard technique described in [19]. The dried shives contained 29.5 wt.% lignin; 41.4 wt.% cellulose; 1.45 wt.% extractives; and 1.9 wt.% ash. Other components (predominantly hemicelluloses) comprised the remaining 25.75 wt% of the material.

A reaction mass was prepared by adding the desired amounts (unless stated otherwise) of the shives (final load of 50 g L$^{-1}$), NaOH solution (final concentration of 50 g L$^{-1}$), water (300 mL), and catalyst solution (CuSO$_4$ · 5H$_2$O; final load of 37.5 g L$^{-1}$) into the reactor while stirring all the while. The catalyst under the process conditions was eventually converted into the form of copper oxide (12 g L$^{-1}$) dispersed on the shives' surface and in the aqueous phase.

The experiments were conducted in two autoclaves of 1 L capacity [19,27]. The reactor for the low stirring intensity (internal diameter: 95 mm) was equipped with a magnetic stirrer (stir bar diameter: 10 mm; stir bar length: 60 mm), whose engine (IKA C-MAG HS 10.8 W) was located under the autoclave's flat bottom. The reactor for the high-intensity stirring (Nano-Mag Technologies, India) was equipped with a six-blade propeller stirrer (diameter: 70 mm) actuated using a 200 W engine via an air-tight magnetic coupling.

The oxidation experiments were conducted at 160 °C and oxygen partial pressure 0.2 MPa at 500 rpm for 60 min, unless specified otherwise. The process conditions were chosen according to our earlier works [19]. Heating (under argon) and temperature stabilization required 25–40 min; then, oxygen was introduced. Oxygen was manually fed into the reactor from a calibrated buffer volume through a valve to maintain the constant pressure in the reactor. The amount of the consumed oxygen was calculated from the pressure change inside the buffer volume. The error of a measurement of the oxygen consumption was determined via the accuracy of the manometer (10 kPa), and it was estimated as 4.2 mmol. Temperature was automatically maintained with a ±2 °C accuracy. Following the oxidation process, the reactor was cooled down to 90 °C for 20 min; then, the excess pressure was vented into the atmosphere; finally, the reactor was opened.

Oxygen consumption rates (mmol/min; Figure 4 and Table 1) were calculated as a tangent of the corresponding line fits (Figure 4) in the interval of 5–20 min.

The majority of the oxidation experiments were performed twice: once to register the oxygen consumption, and once more for sampling the reaction mass to determine the contents of vanillin and syringaldehyde. Samples of the reaction mass (13–15 mL) were taken from the reactor through the special valve during the oxidation process and after the experiment. The aldehyde concentration was determined via GLC [19].

The rate of the reaction mixture homogenization for different volumes of the mixture and different stirring rates was determined at room temperature from the duration of an indicator's color equalization throughout the reactor volume. The indicator (phenolphthalein) was pre-mixed with the reaction mass, and a small amount of alkali was pulse added [32]. A suspension of the flax shives in water was loaded into the reactor, and phenolphthalein was added. Then, the stirrer was turned on, and a small amount (4 mmol) of alkali solution (or 4 mmol of acid into the mass after its alkalinization) was abruptly poured in. The time required for attaining uniform coloration (or complete discoloration in the case of added acid) was determined by recording with a video camera. The homogenization rate was calculated as the inverse of the homogenization time:

$$W_{homogenization} = \frac{n}{N},$$

(12)

where $W_{homogenization}$ is the rate, $n$ is the video frame rate, and $N$ is the number of frames between the addition of the alkali (or the acid) and the coloration equalization.

Experiments on alkali sorption using the shives in different volumes of water at different stirring rates were conducted at room temperature. A total of 2 mmol of NaOH

was added to suspension of 5 wt. % flax shives in water, and the pH change was observed using the MARK-901 pH meter.

The errors of the experimental results were calculated as standard deviations using a least squares method except for those in Table 1 (middle arithmetic errors).

## 4. Conclusions

We studied the influence of mass transfer on the process of powdered flax shive oxidation via molecular oxygen into vanillin, syringaldehyde, and cellulose. The rates of homogenization (equalization of concentrations in the liquid bulk, 0.7–4 s) and of alkali diffusion into the porous structure of the shives (half-life: 26–33 min) were measured at 20 °C. The effect of the addition of active reductants (e.g., ascorbic acid and 1,2,4-triacetoxy benzene) on the rate of oxygen consumption by the reaction mass was studied at 160 °C. The majority of the presented studies were performed under low stirring power density (8 W engine); high-power density was also employed in the autoclave with high stirring intensity (200 W engine).

The stage-wise study of the kinetics of the flax shive oxidation process led to the following conclusions:

(1) The homogenization stage (equalization of reactant concentrations across the liquid bulk) proceeded very fast in terms of characteristic time (1–4 s), and the oxidation reactions took place, for the most part, in this isotropic (homogeneous) aqueous medium.

(2) The internal diffusion of alkali (and by extension, of oxygen) into the porous solid flax shive particles and the diffusion of fragments of the lignin macromolecules outward of the particles proceeded far slower than the liquid homogenization stage, with rates comparable to those of the overall process. However, under low stirring power density, the internal diffusion process did not limit the process rate. In the solid phase at 160 °C, delignification reactions and lignin solubilization occurred under the effects of alkali and oxygen. We believe that the slowest process at this phase boundary is the diffusion of the soluble lignin fragments inside the shive particles towards their outer surface, the external diffusion layer. The transfer from this diffusion layer into the liquid bulk was much faster and does not limit the overall process rate.

(3) The overall rate of the studied process with a low stirring power density (engine power: 8 W) was controlled via the diffusion transfer of oxygen through the diffusion layer at the gas–liquid phase boundary.

(4) At higher mass transfer intensities (engine power: 200 W), the stage of oxygen diffusion through the gas–liquid boundary accelerated and ceased to limit the overall oxidation process rate. However, this process did not enter the chemically controlled kinetic mode; the limiting via the internal diffusion of the reactants and intermediate products inside the particles of the flax shives (or other lignocellulosic materials) set in, whose characteristic time was dozens of minutes, was not affected by the stirring speed. For this reason, a hypothesis can be formulated in that the processes of lignin oxidation by oxygen at 160 °C requiring dozens of minutes exhibit diffusion-controlled kinetics.

(5) Finally, we looked into the question formulated in the Introduction Section regarding the reasons for high selectivity towards the aromatic aldehydes in the diffusion-controlled oxidation of native lignins via oxygen. The theoretical limit of vanillin yield was attained in the three-stage oxidation process of pine-native lignin via molecular oxygen [27]. Attaining this limit indicated that under these conditions, the rates of the lignin depolymerization process, the transfer of the fragments into the bulk solution, and their subsequent oxidation all considerably exceed the rate of the lignin condensation process. This is one reason for the high selectivity.

The second reason is that despite the diffusion limits, the oxidation reaction predominantly takes place in the liquid bulk without concentration gradients of the reactants or products, which can decrease the selectivity of the native lignin oxidation into their

aromatic aldehydes. This state of the reaction mass has been observed over a very wide range of stirring intensities, thanks to the high rate of concentration equalization in the liquid bulk.

Another reason for the high selectivity of the aldehyde formation is that the phenoxyl radical resulting from the single-electron oxidation of the vanillin anion is capable of oxidizing other phenoxide anions (of the remaining lignin), and the overall equilibrium favors the initial anion of vanillin [3].

Therefore, the obtained and discussed results showed that good vanillin yields close to the theoretical limits can be obtained over a very wide range of mass transfer intensities, and it does not matter if the process is limited by the outer diffusion of oxygen through the diffusion layer at the gas–liquid phase boundary or via internal diffusion of the reactants and intermediate products inside the particles of the flax shives (or other lignocellulosic materials).

Developing the study of mass transfer influence on the processes of lignin oxidation via oxygen may permit to enhance its efficiency, to decrease the reagent and energy consumption, and to increase the rate of the process and vanillin concentration in the reaction mass.

**Author Contributions:** Conceptualization, V.E.T. and Y.K.; methodology, V.E.T., A.S.K. and Y.K.; formal analysis, V.E.T., A.S.K. and Y.K.; investigation, K.L.K., M.A.S., Y.V.C.; V.A.G.; data curation, K.L.K., M.A.S., Y.V.C., V.A.G. and V.E.T.; writing—original draft preparation, V.E.T.; writing—review and editing, V.E.T.; visualization, V.E.T., A.S.K., M.A.S., K.L.K. and V.A.G.; supervision, V.E.T. and Y.K.; project administration, V.E.T.; funding acquisition, V.E.T. All authors have read and agreed to the published version of the manuscript.

**Funding:** This research was supported by the Russian Science Foundation, grant No. 20–63-47109.

**Data Availability Statement:** Data are contained within the article.

**Acknowledgments:** The equipment of Krasnoyarsk Regional Research Equipment Centre of SB RAS was used in the experiments. The authors thank Nikolay Tarabanko for their valuable technical assistance.

**Conflicts of Interest:** The authors declare no conflict of interest.

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
