# Peer review of "Mass Transfer in the Processes of Native Lignin Oxidation into Vanillin via Oxygen"

_catalysts, doi:10.3390/catal13121490_

Round 1

Reviewer 1 Report

Comments and Suggestions for Authors

In their paper entitled “Mass Transfer in the Processes of Native Lignin Oxidation into Vanillin by Oxygen”, the authors present their work on the oxidation of an agricultural waste, flax shives to produce vanillin by oxidation. The article is mainly focused on the analysis of the kinetics of vanillin production and oxygen consumption.

The main remarks are:

Studying mass transfer is part of chemical engineering discipline. This science developed so much tools to provide analysis (e.g.: models, dimensionless numbers such as Hatta number) that they should be more used in this article.

Very few information is given about the reaction scheme. Bibliographical references are called, but an article should provide enough information to be fully understood at reading. May the authors add details?

Part of the work is based on the comparison between an 8 W magnetic stirrer and a 200 W stirrer engine. A more interesting figure to be used would the real power dissipated (per volume unit) by the stirrer and not the nominal electric power of the stirrer.

The peculiar remarks are:

L59 “dependeces” should be replaced by dependences.

On figure 1, if O2 external mass transfer would be limiting the O2 concentration in the bulk or in the particle. Where is generated vanillin to have such concentration profile? In the bulk liquid phase?

L103/104 remove “do” in the sentence “may do not”

L140 “in order to fulfil the formulated objectives” would be better.

L174: why not using the same symbol for O2 concentration in equation 3? What is the solubility of O2 in such solutions at such a pressure?

L185: what is the stirring Reynolds number value?

L190: in equation 6 a constant is missing to write the equality between beta and the right part of the equation.

L201: if the results presented are the mean of two experiments, the authors should add the error bars on the graphs. Moreover, in all graph commas should be replaced by dots.

Figures 2 and 3 should have the same abscissas for sake of comparison.

Figure 5: the legend is in the middle of the graph. Comma should be replaced in the legend.

L283 the upsurge could also be due to an inefficient mixing by using a magnetic stirrer.

L314 why using this time the C letter for the constant?

L317: what is the rotation speed for the graph on the right? In the figure 6, the temperature (20°C) should be precised for sake of clarity.

L329: Is the sentence “As far as the intra-particle diffusion is concerned, the latter value may be equal to size of the pores in the flax shives” really true? To describe inner diffusion and reaction, external mass transfer analysis shouldn’t be straightforwardly used.

L356: a double “the the” is present.

Table 3: the amount of ascorbic acid and pyrogallol A are very high. Is the concentration that can be observed during flax shives oxidation? Is the information of figure 7 redundant with table 3?

L424: to intensify the mass transfer, other device (such as gas sparger) could be used instead of vortexes.

L442: “liner”

Figure 10: no real maximum (such as a plateau) can be seen in the data for 500 rpm. Why the vanillin yield is not equal to 0 at the start?

Figure 11: the thickness of the diffusion layer should be reduced at high stirring speed.

Comments on the Quality of English Language

There are a few typos in the article.

Author Response

Reviewer Remarks_19_11_2023

Responses 2.

Dear Reviewer,

thank you very much ones more.

I am sorry for the non-correct understanding your remarks.

I corrected the manuscript according to your remarks.

Thank you ones more.

Valery E. Tarabanko

Rev.  1.

Dear Reviewer,

Great thanks for the fruitful remarks.

We revised the manuscript according to your remarks. Our responses are written below, and corresponding corrections of the text are highlight in yellow colour.  

In their paper entitled “Mass Transfer in the Processes of Native Lignin Oxidation into Vanillin by Oxygen”, the authors present their work on the oxidation of an agricultural waste, flax shives to produce vanillin by oxidation. The article is mainly focused on the analysis of the kinetics of vanillin production and oxygen consumption.

The main remarks are:

Studying mass transfer is part of chemical engineering discipline. This science developed so much tools to provide analysis (e.g.: models, dimensionless numbers such as Hatta number) that they should be more used in this article.

You are absolutely right in this remark.

But the paper deals with quiet complicated system. It consists of four phases: liquid water, gaseous oxygen, catalyst, and solid particles of flax shives. The solid particles change their size and reactivity during the oxidation, starting from 1 mm size down to micronized pulp particles, and this is topochemical process with own unknown regularities. 

As for the rate of the chemical reaction, we cannot estimate its rate constant and, hence, Hatta number due to several reasons. The main reaction determined the total rate of O2 consumption is probably the reaction of oxygen with soluble destructed lignin in a bulk water phase. Soluble oligomeric destructed lignin is formed by reactions of solid lignin of lignocellulose with alkali and oxygen. This rate determined by internal diffusion is probably much less compared to the registered total rate of oxygen consumption.  

As a result, we cannot estimate a quasi-stationary concentration of soluble forms of lignin in solution during the process of oxydation.    

Next problem of total oxygen consumption rate and the rate of vanillin formation is that these rates differ by 20-30 times due to deep oxidation of lignin and soluble carbohydrates in the solution.

These are remarks on the complicated chemistry of the process under study.

Similarly, there are many problems with description of mass transfer in the process.

The first is estimation of viscosity of the reaction mass like at room temperature and moreover under the process conditions.

We tried to measure viscosity of two-phase system water-flax shives by capillary (diameter of 8 mm) and falling-sphere viscometers, but without any successes. The results of our previous paper [BCBR] indicated this is non-Newtonian suspension, and studying its viscosity is a special task outside the framework of the manuscript. 

We do not know also a gas bubble quantity in the suspension under the conditions in the reactor, at high pressure and temperature.

On these reasons dimensionless numbers (including Reynolds number, see our response on your L185 remarks) seems to be impossible to calculate and to use for describing the process based on the experimental results obtained.  

This and previous [] our papers are the first which connected the rate of the process under study (oxidation of lingocellulose by oxygen into vanillin) with the mass transfer conditions in the reactors. No quantitative models to describe these processes are presented in literature.

On these reasons, we used a simplest and well-known model (two calculated coefficients) to describe the kinetic data obtained.  

We cannot measure a  torque (Newton-meter) of stirrer now and before, but in the nearest future we will build an installation for this measurement at normal pressure. This permits us to measure stirring power density and even viscosity of the reaction mass, suspension. Hence, we will be able to estimate some dimensionless numbers and to apply more complicated models for describing the process.    

Very few information is given about the reaction scheme. Bibliographical references are called, but an article should provide enough information to be fully understood at reading. May the authors add details?

Dear Reviewer, you are right, and I realized it clearly when writing response on your previous remark. 

We inserted the next paragraphs into the introduction.

While studying mass transfer in the process of lignocellulose oxidation by oxygen several complications and difficulties should be taken into account.

Lignin irregular structure is formed as a result of oxidative dehydrogenation of coniferyl, synapyl, and p-coumaric alcohols.  Lignins are kinetically inhomogeneous during oxidation into vanillin, and in the first approximation it may be considered as consisted of two fractions (Fig. 1) [3]. The first contains more β-O-4 bonds between coniferyl units (structure (I)), and the second fraction (II) includes more 5-5’ bonds. The first structure can produce vanillin during oxidation,

and the second cannot.

Figure 1. The most important structures of coniferous lignin.

The process of lignin oxidation includes four phases: liquid water solution of alkali, gaseous oxygen, catalyst, and solid particles of flax shives. The latter solid particles change their size and reactivity during the oxidation, starting from 1 mm size down to micronized pulp particles, and this is topochemical process with own unknown regularities. Ones more, lignin is irregular polymer (Fig. 1) consisted of different structures with different activities.  

First stage of the process is delignification, destruction and dissolution of solid lignin as a result of its heterogeneous reaction with alkali and solubilized oxygen (Fig. 2). Internal diffusion of reagents and products of the stage should be taken into account while discussing its role in the process in total. Probably, oxygen consumption in this stage is relatively slow compared to the total oxygen consumption in the process.

Figure 2. Main stages of the process of lignin oxidation

It should be noted a zero stage of the process: dissolution of a most reactive part of lignin in the alkaline solution during heating the reactor without oxygen. A part of vanillin is formed in this stage as a result of retroaldol reaction of substituted coniferyl aldehyde. According to our viewpoint, all vanillin is obtained due to such retroaldol reaction, and oxygen is necessary in the process to oxidize substituted coniferyl alcohols into the aldehyde.  

Second (and the main) stage of the process is catalytic and non-catalytic oxidation of oligomers of destructed lignin. Main part of vanillin is obtained in this stage, and the most part of oxygen consumed here. The total oxygen consumption exceeds the vanillin yield (mol per mol) in the process by a factor of 20-30 due to deep oxidation of lignin and soluble carbohydrates in the solution.  The obtained carboxylic acids and CO2 consume analogous quantity of alkali. 

As for the rate of the chemical reaction, we cannot estimate its rate constant and, for example, Hatta number due to several reasons. The main reaction determined the total rate of O2 consumption is probably the reaction of oxygen with soluble destructed lignin in a bulk water phase. Soluble oligomeric destructed lignin is formed by reactions of solid lignin of lignocellulose with alkali and oxygen. This rate determined by internal diffusion is probably much less compared to the registered total rate of oxygen consumption. As a result, we cannot estimate a quasi-stationary concentration of soluble forms of lignin in solution during the process of oxidation, which is necessary to estimate Hatta number.

The third (and undesirable) stage is vanillin oxidation into vanillic acid and, finally, CO2, Na2CO3. Vanillin as phenol is relatively more stable compared to phenolic groups of lignin under the conditions of oxidation due to electron-accepting properties of the carbonyl p-substituent.

All these three consecutive stages except the zero one occur simultaneously and to separate the kinetic of certain stage is a special and complicated task.  

The second part of this clarifications are inserted after equation (7).  

The complication of chemistry of the process under study was discussed above (Fig. 1, 2). Similarly, there are many problems with description of mass transfer in the process. The first is estimation of viscosity of the reaction mass like at room temperature and moreover under the process conditions.

We tried to measure viscosity of two-phase system water-flax shives by capillary (diameter of 8 mm) and falling-sphere viscometers, but without any successes. The results of our previous paper [19] indicated this is non-Newtonian suspension, and studying its viscosity is a special task outside the framework of the manuscript. 

Therefore, dimensionless numbers (including Reynolds number) seems to be impossible to calculate and to use for describing the process based on the experimental results obtained. On these reasons, we used a simplest and well-known model (two calculated coefficients, Equation 7) to describe the kinetic data obtained.  

Part of the work is based on the comparison between an 8 W magnetic stirrer and a 200 W stirrer engine. A more interesting figure to be used would the real power dissipated (per volume unit) by the stirrer and not the nominal electric power of the stirrer.

Dear Reviewer, I agree with you completely. I noted above that we are to make the installation for measuring the real power dissipated (per volume unit). All we were able to make when preparing the manuscript was to compare geometry of funnels for different stirrers and engines (Fig. 8, Fig. 10 in the revised manuscript).

We may assume that equalled funnel geometry means the same Froude number Fr = n2dstirrer/g, and taking into account the same height of the funnels at double differs of stirring rates for different engines we may conclude that the real power dissipated (per volume unit) differs by 4 times for the engines of 8 and 200 W at the same stirring rate. 

To conclude my responses, I believe that the experimental results presented in the manuscript cannot be convincingly described by any models or dimensionless numbers except the applied model of power dissipated (per volume unit) which determines the process rate. But this manuscript is the first attempt to describe quantitatively mass transfer processes in the field of research, oxidation of lignins into vanillin by oxygen. Therefore, I believe the paper worth publishing.  

The peculiar remarks are:

L59 “dependeces” should be replaced by dependences. - done

On figure 1, if O2 external mass transfer would be limiting the O2 concentration in the bulk or in the particle. Where is generated vanillin to have such concentration profile? In the bulk liquid phase?

Yes, in the bulk liquid phase, and it is noted in the comments to Fig. 2 of corrections.  .

L103/104 remove “do” in the sentence “may do not” - done

L140 “in order to fulfil the formulated objectives” would be better. - done

L174: why not using the same symbol for O2 concentration in equation 3? - done

What is the solubility of O2 in such solutions at such a pressure?

Extrapolation of the data on O2 solubility in 1M NaOH at 373 K, 1 Bar (0.51 mmol/L) [Wei Xing, Min Yin, Qing Lv, Yang Hu, Changpeng Liu, Jiujun Zhang 1 - Oxygen Solubility, Diffusion Coefficient, and Solution Viscosity. In: Rotating Electrode Methods and Oxygen Reduction Electrocatalysts. Edited by: Wei Xing, Geping Yin and Jiujun Zhang. 2014 Elsevier B.V. Pages 1-31. http://dx.doi.org/10.1016/B978-0-444-63278-4.00001-X]  to 160 °C and 2 Bar gives O2 solubility (Csat) of appr. 0.8 mmol/L. This datum included into the manuscript.

 A simple and natural explanation for this result can be found in the model that links the rate of a chemical process limited by mass transfer to the stirring power density [19-23,31,32]. The simplest description of mass transfer rate in a gas-liquid system (which corresponds to the oxygen consumption rate in our case) is provided by two Equations (3), (4) [32]:

 mol m-3s-1                                         (3)

where q is oxygen consumption,  is volumetric coefficient of mass transfer (s−1), Csat. and C are the saturation and current aqueous concentrations of oxygen respectively. Extrapolation of the data on O2 solubility in 1M NaOH at 373 K, 1 Bar (0.51 mmol/L) [33] to 160 °C and 2 Bar gives O2 solubility (Csat) of appr. 0.8 mmol/L.

L185: what is the stirring Reynolds number value?

Using equation Re = nd2stirrer · r/μ,  where n = 15 – stirring rate, sec-1 , dstirrer  = 0.08 m – stirrer diameter, r = 1000 kg/m3 – reaction mass density,   μ = 0.001-1 Pa · sec – dynamic viscosity (possible interval, we tried but were not able to measure the viscosity of our suspension) gives the Re estimation Re ~ 100-100 000. So, we cannot even conclude formally if there is laminar (Re < 2700) or turbulent (Re > 104) mode near the agitator. Using equation Re = wl/μ does not simplify the problem.

L190: in equation 6 a constant is missing to write the equality between beta and the right part of the equation. – thank you, const is inserted in the equation.

L201: if the results presented are the mean of two experiments, the authors should add the error bars on the graphs. Moreover, in all graph commas should be replaced by dots. Thank you, done.

Figures 2 and 3 should have the same abscissas for sake of comparison. Thank you, done.

Figure 5: the legend is in the middle of the graph. Comma should be replaced in the legend. Thank you, corrected.

L283 the upsurge could also be due to an inefficient mixing by using a magnetic stirrer.

Yes, you are right, a low power of the magnet stirrer may be not enough to provide Newtonian behavior of large volumes of the reaction mass. We corrected the sentence.

There is a noteworthy rift between homogenization rates in liquid volumes 200 – 300 mL and 400 – 600 mL. This rate upsurge may be caused by non-Newtonian behavior of the viscous suspension where its viscosity increases with lowering gradient of the fluid motion speed at greater distances from the stirrer bar because of low stirring power [19].

L314 why using this time the C letter for the constant? Thank you, C changed by const.

L317: what is the rotation speed for the graph on the right? In the figure 6, the temperature (20°C) should be precised for sake of clarity. Thank you, done.

L329: Is the sentence “As far as the intra-particle diffusion is concerned, the latter value may be equal to size of the pores in the flax shives” really true? To describe inner diffusion and reaction, external mass transfer analysis shouldn’t be straightforwardly used.

The sentence is not correct. Diffusion coefficients in large pours and in diffusion layers are the same.  We deleted the sentence.

L356: a double “the the” is present. Thank you, done.

Table 3: the amount of ascorbic acid and pyrogallol A are very high. Is the concentration that can be observed during flax shives oxidation?

Yes, we used high concentrations to observe increasing the oxygen consumption rate at the initial moment of the reaction (t ~ 1 min).

 Is the information of figure 7 redundant with table 3? Table 3 repeats partially the data of Fig. 7, and we deleted a part of the Table.

L424: to intensify the mass transfer, other device (such as gas sparger) could be used instead of vortexes.

Yes, thank you, we are to use different methods of mass transfer intensification, but such approaches are outside the frame of the study.

L442: “liner”  Thank you, linear,  done.

Figure 10: no real maximum (such as a plateau) can be seen in the data for 500 rpm.

Yes, you are right. We added one more dot (it’s the curve from Fig. 3 (5 in revised vertion)). Really, maximum is not observed, but our numerous results [19] show this is really maximum. 

Why the vanillin yield is not equal to 0 at the start?

A part of vanillin is formed during heating the reactor (20-160 °C) without oxygen. We noted it in the expanded Introduction.

Figure 11: the thickness of the diffusion layer should be reduced at high stirring speed. Thank you, we corrected the Figure. 

Comments on the Quality of English Language

There are a few typos in the article. Thank you for the corrections.

Dear Reviewer, thank you very, very much for the fruitful reviewing the manuscript.

Reviewer 2 Report

Comments and Suggestions for Authors The present manuscript deals with the study of the influence of mass transfer intensity on the kinetics of catalytic oxidation of flax shives with oxygen in alkaline media to aromatic aldehydes and pulp was studied. The process was carried out in two autoclaves, with moderate stirring (stirrer engine of 8 and 200 W).   The Introduction reported an exhaustive survey of the status of the art of this research theme.   The results and discussion is satisfactory although there are minor changes to be done. 1) sign + in [H+] must be reported in superscript. In aqueous solution H3O+ must be preferred instead of H+. 2) please, explain the meaning of the symbol ln (is it the natural logarithm?) 3) line 313: k/ln10 is not "the plot slope in Figure 6", but "the slope of the regression line obtained by plotting pH vs. time.". 4) Data added in the text with decimals must be always reported with dot, rather than with comma. Please, revise accordingly.   Conclusions are not simply a summary of the results obtained. I suggest also the authors add future perspectives and to stress mainly relevant results and why they are so important.    For all the above mentioned reasons I recommend the publication of this manuscript after minor revision. Comments on the Quality of English Language

minor editing of the English is needed

Author Response

Rev. 2.

Dear Reviewer,

Great thanks for the fruitful remarks.

We revised the manuscript according to your remarks. Our responses are written below, and corresponding corrections of the text are highlight in yellow colour. 

The present manuscript deals with the study of the influence of mass transfer intensity on the kinetics of catalytic oxidation of flax shives with oxygen in alkaline media to aromatic aldehydes and pulp was studied. The process was carried out in two autoclaves, with moderate stirring (stirrer engine of 8 and 200 W).   The Introduction reported an exhaustive survey of the status of the art of this research theme.  

The results and discussion is satisfactory although there are minor changes to be done.

1) sign + in [H+] must be reported in superscript. In aqueous solution H3O+ must be preferred instead of H+. - Done

2) please, explain the meaning of the symbol ln (is it the natural logarithm?) – yes, it is the natural logarithm, it is noted in the text.

3) line 313: k/ln10 is not "the plot slope in Figure 6", but "the slope of the regression line obtained by plotting pH vs. time.". – done.

4) Data added in the text with decimals must be always reported with dot, rather than with comma. Please, revise accordingly.   – done.

Conclusions are not simply a summary of the results obtained. I suggest also the authors add future perspectives and to stress mainly relevant results and why they are so important.   

Yes, thank you, we added two phrases to the conclusion.

So, the obtained and discussed results show that good vanillin yields closed to the theoretical limits can be obtained in a very wide range of mass transfer intensity, and it does not matter if the process is limited by outer diffusion of oxygen through the diffusion layer at the gas-liquid phase boundary or by internal diffusion of the reactants and intermediate products inside the particles of the flax shives (or other lignocellulosic materials).

Developing the study of mass transfer influence on the processes of lignins oxidation by oxygen may permit to enhance its efficiency: to decrease the reagents and energy consumption, to increase the rate of the process and vanillin concentration in the reaction mass. 

For all the above mentioned reasons I recommend the publication of this manuscript after minor revision.

Comments on the Quality of English Language

minor editing of the English is needed
